# Chemorefractory Gastric Cancer: The Evolving Terrain of Third-Line Therapy and Beyond

**DOI:** 10.3390/cancers14061408

**Published:** 2022-03-10

**Authors:** Maria Alsina, Josep Tabernero, Marc Diez

**Affiliations:** 1Gastrointestinal & Endocrine Tumours Group, Vall d’Hebron Institute of Oncology (VHIO), C/ Natzaret 115-117, 08035 Barcelona, Spain; malsina@vhio.net (M.A.); mdiez@vhio.net (M.D.); 2Medical Oncology Department, Hospital Universitario de Navarra (HUN), Universidad Pública de Navarra (UPNA), Irunlarrea 3, 31008 Pamplona, Spain; 3Medical Oncology Department, Hospital Universitari Vall d’Hebron (HUVH), Universitat Autònoma de Barcelona (UAB), Pg. Vall d’Hebron 119-129, 08035 Barcelona, Spain

**Keywords:** gastric cancer, gastro-oesophageal junction cancer, third line of treatment, sequential treatment approach, molecular approach

## Abstract

**Simple Summary:**

Gastric and gastro-oesophageal junction cancers (GC) are the fourth cause of cancer-related deaths, representing an international problem which needs a proper assessment. Beyond being an aggressive disease, which rapidly progress to different lines of treatment, patients suffering GC normally present a non-depreciable number of symptoms that make them especially fragile. In this context, sequential treatment lines with few toxic adverse events have been associated to an increased survival. A satisfactory GC therapy comprise at least three lines of treatment including chemotherapy and immunotherapy and/or targeted agents when indicated.

**Abstract:**

Gastric and gastro-oesophageal junction cancer (GC) represent a global healthcare problem being the fifth most common tumour type and the fourth cause of cancer mortality. Extremely poor median survival of approximately 10 months is normally reported within advanced GC patients, mainly secondary to two factors, i.e., the fragility of these patients and the aggressiveness of this disease. In this context, the correct treatment of GC patients requires not only a multidisciplinary team with special attention to palliative and nutritional care but also a close follow-up with regular monitoring of disease symptoms and tumour evaluation. Sequential treatment lines with few toxic adverse events have emerged as the best therapeutic approach, and a third line of therapy could further improve survival and quality of life of GC patients. Chemotherapy, immunotherapy, and targeted agents -when indicated- constitute the treatment armamentarium of these patients. In this review, we discuss treatment options in the refractory setting as well as novel approaches to overcome the poor prognosis of GC.

## 1. Introduction

Gastric and gastro-oesophageal junction cancer (GC) constitutes a global healthcare problem due to its high prevalence and aggressiveness. GC ranks fifth in cancer incidence and fourth in cancer mortality [1].

Patients with advanced GC have an extremely poor prognosis with a median overall survival (OS) of 3 months when treated only with best supportive care [2]. A first chemotherapy line has been demonstrated to improve OS up to 10 months, together with betterment of quality of life (QoL) [2] and should be offered in patients presenting an adequate performance status and organ function [3]. GC patients are very fragile, and then the best chemotherapy approach lies on the sequentially of treatment lines with mild toxic adverse events, together with close monitoring with frequent imaging assessments.

GC patients have been historically treated as they had a unique disease, without recognizing its inter- and intratumoural intrinsic heterogeneity, except the targeted approach to the HER2-positive population and some immunological strategies.

The inter-patient heterogeneity was initially described during the first decade of the 21st century [4,5] and definitively characterised in 2014 [6] when The Cancer Genome Atlas (TCGA) analysis of GC comprehended four molecular subtypes: Epstein-Barr virus (EBV), microsatellite instability (MSI), chromosomal instability (CIN) and genomically stable (GS), all of them represented along the stomach, although differently. The EBV and MSI subtypes show the highest immune infiltration. The EBV tumours are infiltrated by high levels of immune cells with positivity of PD-L1 and PD-L2, secondary to the virus infection. In contrast, the lymphocyte infiltration of MSI tumours is due to an impaired DNA mismatch repair (MMR) function [7]. The most frequent subtype is the CIN subtype (50%), which is mostly represented in the proximal stomach and has also been described in the gastro-oesophageal junction and distal oesophagus. The CIN subtype is characterised by frequent *TP53* mutations and amplifications of the genes responsible of the receptor tyrosine kinase proteins (RTKs, such *EGFR*, *ERBB2 (HER2)*, *ERBB3*, *FGFR2* and *MET*), *KRAS* or *NRAS*, cell-cycle mediators and *VEGFA.* Finally, the GS subtype, mostly represented in the distal stomach, enriched by the diffuse histology and with *CDH1* or *RHOA* mutations, or the *CLD18-ARHGAP26/6* translocation. In line with the TCGA molecular classification, the Asian Cancer Research Group (ACRG) [8] published a similar molecular stratification at the same time. Altogether, these findings support the potential of GC to be treated with precision oncology strategies. 

Although the well described role of Helicobacter pylori infection contributing to the non-cardia carcinogenesis [9], no convincing evidence has been attributed to the gastric microbiome. Indeed, the damage caused by the chronic inflammation secondary to the bacterial infection would be the major cause of the cancer pathogenesis [10,11,12].

GC heterogeneity has been the major reason of the recurrent failure of biomarker and non-biomarker-based clinical trials [13,14,15,16,17,18,19,20,21]. GCs heterogeneity relies on differences between patients’ tumours (inter-patient variability) but also within the same patient (intratumoural heterogeneity) [22,23]. The last including a spatial and a temporal concept, with differences found between the primary tumour and the metastasis [22]. Actually, whether discordances between the primary lesion and the recurrence have been found in up to 36% of cancers, a concordance between the metastasis and ctDNA has been mostly found (87.5%) [24]. In addition, acquired molecular alterations such as *EGFR* amplifications, *FGFR2* fusions and *MET* amplifications have been identified as resistance mechanisms to directed target therapies [25,26].

## 2. Initial Approach

Five types of chemotherapy drugs have demonstrated activity in GC: platinum compounds, fluoropyrimidines (5-fluorouracil, capecitabine and S-1), taxanes, irinotecan and trifluridine-tipiracil [27,28,29]. The typical initial approach of the metastatic disease includes a first line based on a platinum and a fluoropyrimidine doublet [27]. The addition of trastuzumab in those patients with HER2-positive tumours [HER2 immunohistochemistry (IHC) score 3+, or HER2 IHC 2+ and fluorescent in situ hybridisation (FISH/ISH)-positive] is required, secondary to the demonstrated survival benefit in the TOGA Trial [30]. Finally, the inclusion of an immune checkpoint inhibitor (ICI) is necessary in those patients with tumours exhibiting high programmed death ligand 1 (PD-L1) combined positive score (CPS), in consonance with the recent benefit demonstrated by the programmed cell death protein 1 (PD-1) nivolumab and pembrolizumab [31,32] (Figure 1).

Those GC patients who arrive fit enough for a second line should be treated with a taxane or irinotecan. When possible, the combination of paclitaxel and ramucirumab should be offered, as it is the treatment with the best academic support [33].

To date, no other targeted agent has demonstrated a survival benefit in this setting, probably due to an insufficient biomarker trial stratification, in addition to the described intrinsic GC heterogeneity.

## 3. Close Follow-Up

Patients’ expectations, comorbidities, organ function and performance status (PS) should always be considered when selecting any treatment regimen, especially for GC patients due to high patient fragility and disease aggressiveness.

GC patients normally debut with a median of 65-years of age, with implied comorbidities and the presence of non-depreciable symptomatic disorders. Dysphagia or dyspepsia and vomiting are frequently reported depending on tumour localization, together with some dull aching pain secondary to tumour or metastatic lesions. Finally, anaemia and malnutrition are often associated. The everyday maintenance of supportive care needs a multidisciplinary approach. The preservation of the PS includes not only the assurance of an adequate organ function and symptomatic control, but also a good nutritional status. Both the tumour location into the stomach and the aggressiveness of this disease contribute to the frequently associated GC cachexia. Cancer cachexia is an inflammatory condition that clearly correlates with a poor prognosis [34,35]. To date, no anti-inflammatory treatment has been sufficiently correlated with the control of the sustained cancer-induced cachexia, only nutritional support being the best effective way to manage it [36,37]. Nutritional support should be provided at disease debut and throughout routine evaluations.

Most GC patients present with inoperable advanced or metastatic disease at diagnosis, and approximately 50% of operated patients develop recurrence after surgery [38]. When treated with a palliative intention, progression-free survival (PFS) times are short, i.e., around 6 months in the first line [30,31] and 4 months in the second line [33].

Even considering these two intrinsic factors, patients’ fragility and the aggressiveness of the disease, real-world data studies have extensively demonstrated a survival benefit when patients are able to receive more treatment lines [39,40,41], although the proportion of patients that receive subsequent treatment lines progressively decreases [40,41,42]. In the frail elderly population, un upfront dose-reduction (80–60%) of chemotherapy using lower intensity regimens does not compromise the tumour and symptom control, thus probably being a better option [43].

To summarise, an appropriate approach to respond to GC patients’ needs requires (1) a regular evaluation and treatment by a multidisciplinary team in order to improve the tumour burden and maintain patient PS and quality of life, and (2) an image monitoring close follow-up performed every 2–3 months. If radiological progression is anticipated prior to the patients’ PS deterioration, then these patients would be able to receive the optimal treatment sequence.

Finally, treating physicians are recommended to consider the effectiveness of each treatment line before choosing the following one. A positive correlation has been demonstrated between effectiveness of the first-line therapy with outcomes of the second-line therapy [44], which could probably be translated into the consecutives lines.

## 4. Clinical Trials in the Refractory Setting

### 4.1. Chemotherapy

Despite the lack of randomised phase III trials, treatment with taxanes or irinotecan have been positioned as acceptable for the third line. In this setting, the TAGS Trial demonstrated the value of trifluridine/tipiracil when compared with the placebo, making this drug useful in these countries with reimbursement [45] (Figure 1 and Table 1).

Trifluridine/tipiracil is an oral chemotherapy agent composed by a nucleoside analogue, i.e., trifluridine, a thymidine-base, and tipiracil, a thymidine phosphorylase inhibitor. The antitumour activity mainly relies on the inhibition of the cell proliferation secondary to the incorporation of the trifluridine within the replicating DNA strands. Moreover, and although not as important, trifluridine also inhibits the thymidine synthetase (TS), an enzyme necessary for DNA synthesis. On the other hand, tipiracil inhibits the thymidine phosphorylase (TP), thus inhibiting the trifluridine degradation and consequently increasing its availability.

In the TAGS trial, 507 GC patient’s refractory to at least two lines of chemotherapy were randomised 2:1 to receive trifluridine/tipiracil plus best supportive care vs. placebo plus best supportive care [45] (Table 1). Patients treated with trifluridine/tipiracil had a median overall survival (OS) of 5.7 months, compared with 3.6 months in patients receiving placebo [hazard ratio (HR) 0.69 [95% confidence interval (CI) 0.56–0.85; *p* < 0.001]. The survival efficacy was sustained independently of different prognostic factors as ECOG/performance status, number of previous treatment lines, HER2 status, number of metastatic sites and age. Although the response rate (ORR; 4%) was relatively low, trifluridine/tipiracil offered a good disease control rate (DCR) (DCR;44%). Most of the trifluridine/tipiracil adverse events (AEs) were non-complicated haematological toxicities, and the quality of life (QoL) of the patients was maintained similarly for both groups, within the different functional and symptom scales. Indeed, a positive trend toward a better QoL was observed within patients receiving trifluridine/tipiracil [46].

Apart from chemotherapy options, immune strategies and some targeted therapies have been tested in the 3rd line and refractory setting.

### 4.2. Immunotherapy

Although treatment with ICIs has demonstrated activity in the chemorefractory setting [47,48], they would be translated to the first line in those patients with tumours expressing PD-L1 CPS ≥ 5, given the demonstrated survival benefit [31,49] (Table 1).

Nivolumab demonstrated OS superiority over placebo in the phase III randomised Asian trial, in refractory patients (progression to ≥2 prior lines) [47]. In this trial, 493 GC patients from Japan, South Korea and Taiwan were randomised 2:1 to receive nivolumab 3 mg/kg biweekly or placebo. Patients were refractory to at least 2 prior lines, although 40% had received 3, and 40% ≥ 4 prior lines. Most of patient had GC localized in the stomach (82%). The primary end point was met, with 5.26 months of OS in the nivolumab group vs. 4.14 months in the placebo group [HR 0.63 (95% CI 0.51–0.78), *p* < 0.001]. The ORR in the nivolumab group was 11.2% (95% CI 7.7–15.6). Treatment with nivolumab was well tolerated, with grade 3 or 4 treatment-related adverse events occurring in 10% of patients who received nivolumab (vs. 4% in the placebo arm).

Pembrolizumab showed similar efficacy in an international phase II trial [48]. In cohort 1 of this phase II trial 259 refractory GC patients from US, East Asia and other sites were treated with pembrolizumab 200 mg three-weekly. A total of 48% had gastric cancer and 51% gastro-oesophageal junction cancer. The 29% were refractory to 3 lines, whereas a 19% to ≥ 4 lines of previous treatment. The OS was 5.5 months (95% CI 4.2–6.5), and the ORR was 11.6% (95% CI 8.0–16.1). The ORR was higher in patients with tumours expressing PD-L1 CPS ≥1, assessed by the pharmaDx-223 IHC assay [ORR of 15.5% (95% CI 10.1–22.4%)]. Based on these data, the U.S. FDA approved pembrolizumab for GC patients with tumours expressing PD-L1 CPS ≥ 1 and after progression to at least two treatment lines.

Unfortunately, the anti-PD-L1 antibody avelumab did not demonstrate any OS superiority when compared with chemotherapy, in the 3rd line setting, although clinical benefit was seen in the PD-L1 CPS ≥ 1 population [50]. This phase III trial randomised 1:1 371 GC patients from Europe, Asia and North America receive avelumab 10 mg/kg biweekly or physicians’ choice chemotherapy (paclitaxel or irinotecan). The primary end point of OS was not met [mOS 4.6 vs. 5.0 months; HR 1.1 (95% CI 0.9–1.4); *p* = 0.81], neither the secondary end point of PFS [mPFS 1.4 vs. 2.7 months; HR 1.73 (95% CI 1.4–2.2); *p* > 0.99], nor ORR (2.2% vs. 4.3%) in the avelumab versus chemotherapy arms, respectively.

The validation of the other biomarkers of immune response, MSI-H/dMMR and TMB-high, was based on phase II/III studies conducted specifically in GC but also in other solid tumours [51,52]; and the U.S. FDA has approved pembrolizumab in metastatic GC patients with MSI-H/dMMR and TMB-high (≥10 mut/mb) tumours.

These encouraging results opened the door to explore novel combinations, which are currently being evaluated in different clinical trials. As an example, some of them are trying to demonstrate the thoughtful synergistic association of ICIs plus multikinase inhibitors (TKIs), which theoretically mitigate the PD-1 resistance by inhibiting T reg cells [53,54]. Actually, primary signs of efficacy have already been shown with the combination of regorafenib plus nivolumab and of lenvatinib plus pembrolizumab [55,56].

### 4.3. Target Therapies

Different target therapies have been tested for this refractory population, i.e., anti-angiogenic strategies with apatinib [57,58], mammalian target of rapamycin (mTOR) inhibition with everolimus [59] and anti-HER2 blockade with the antibody-drug conjugate (ADC) trastuzumab deruxtecan (T-DXd) [60] (Table 1).

Apatinib, a TKI targeting the vascular endothelial growth factor-2 (VEGFR2), improved OS versus placebo in refractory GC in a phase III study conducted in China [57], leading to its approval by the Chinese FDA. In this case, 267 GC patients were randomised to oral apatinib 850 mg or placebo once daily. A total of 70% of patients had gastric cancer, the rest (30%) gastro-oesophageal junction cancer. Of them, the majority had progressed to 2 previous lines (65%) or to ≥3 (35%). The primary end point mOS was significantly improved in patients receiving apatinib compared with the placebo group [6.5 vs. 4.7 months; HR 0.71 (95% CI 0.54–0.94); *p* = 0.015]. Unfortunately, the international phase III randomised ANGEL trial did not validate these results [58]. The ANGEL study randomised 460 GC patients to receive 2:1 rivoceranib (apatinib) 700 mg or placebo once daily. Stratification factors included the geographic region (Asian vs non-Asian), disease measurability, prior treatment with ramucirumab, and number of previous lines). The OS was not significantly different between the two groups. [5.78 vs. 5.13 months; HR 0.93 (95% CI 0.74–1.15); *p* = 0.485].

On the other hand, everolimus was tested in the international phase III GRANITE-1 study, for 2nd and 3rd line of treatment, but with poor results [59]. This study randomised 2:1 656 GC patients to receive everolimus 10 mg/d or placebo. Stratification was by number of previous chemotherapy lines (one vs. two) and region (Asia vs. rest of the world). The primary end point mOS was 5.4 months vs. 4.3 months in patients treated with everolimus vs placebo, respectively [HR 0.90 (95% CI 0.75–1.08), *p* = 0.124].

It is important to mention that none of these above-related studies selected the GC patients by any biomarker, which probably explains the negative results.

The historical success has come from studies targeting the HER2-positive GC population. The phase II DESTINY-Gastric01 trial evaluated the treatment with T-DXd in the HER2 setting, demonstrating a dramatic benefit in this context. T-DXd is an antibody-drug conjugate with a cytotoxic topoisomerase I inhibitor [60] that potentially attacks not only the HER2-positive cells but also the other surrounding ones by a bystander effect. In the DESTINY-Gastric01 trial, 187 Asian HER2-positive GC patients who had received at least two previous lines of therapy, including trastuzumab, were randomised 2:1 to receive T-DXd 6.4 mg/kg every 3 weeks or physician’s choice of chemotherapy. Both, the ORR and the mOS were improved within patients receiving T-DXd compared with patients receiving placebo: ORR of 51% vs. 14% (*p* < 0.001), and mOS 12.5 vs. 8.4 months [HR 0.59 (95% CI 0.39–0.88; *p* = 0.01] [60]. The safety profile of T-DXd was acceptable, with most of the grade ≥ 3 AEs being haematological, except for interstitial lung disease or pneumonitis related with T-DXd, reported in 12 patients (10%; grade 1-2 in 10 patients and grade 3-4 in 3). Interestingly, and secondary to this bystander capacity, T-DXd also showed antitumour activity in HER-2 low expressing patients (IHC 2+ and ISH negative or IHC 1+) [61].

Furthermore, the primary analysis of phase II single-arm trial of T-DXd in western patients (Belgium, Great Britain, Italy and Spain) was positively reported [62]. A total of 79 GC patients with centrally confirmed HER2 positive disease on biopsy after progression to first-line trastuzumab-containing regimen were treated with T-DXd at 6.4 mg/kg every 3 weeks. Most of them had gastro-oesophageal junction cancer (65.8%) and had progressed to ≥2 treatment lines. Confirmed ORR was 38% with a mPFS of 5.5 months (95% CI 4.2–7.3), thus demonstrating clinically meaningful activity. Drug-related treatment-emergent adverse events grade ≥3 were reported in 27%, although only 8.9% were associated with treatment discontinuation [investigator-reported pneumonitis (3.8%) and interstitial lung disease (2.5%)]. These positive results support the ongoing randomised phase III trial (DESTINY-Gastric04; NCT04704934).

Although T-Dxd is currently being considered for the 2nd line treatment of the HER2-positive population, some combinations of this antibody with ICIs in the refractory setting are under evaluation [63]. Likewise, the synergism of the anti-HER2 and anti-PD1 combination strategies has also been reported in the phase 1-2b trial with the anti-HER2 antibody margetuximab and pembrolizumab [64].

### 4.4. Novel Treatment Approaches

Fortunately, the approach of GC patients will improve in the future, considering the increasingly expanded use of sequencing technologies in the daily clinical setting. This approach, instead of sequentially testing of unique molecular alterations, offers the opportunity of drawing a specific “molecular” algorithm from the beginning of the disease. First, the precise biomarker selection would probably translate in a more effective treatment. Second, doing it one time at the debut of the disease would accelerate treatment decisions, which is crucial in for this aggressive disease. As an example, the VIKTORY and the PANGEA trials. 

The VIKTORY trial [65] was an umbrella trail conducted in South Korea. This study integrated 10 phase II clinical trials targeting 8 biomarkers, for GC patients that had progressed to a first line of treatment. The molecular alterations included *RAS* aberrations, *TP53* mutations, *PIK3CA* mutations and/or amplifications, *MET* amplifications, MET over-expressions, all negative, *TSC2* deficiency and *RICTOR* amplifications. This study demonstrated a prolonged PFS and OS in those patients receiving the biomarker-selected therapy, suggesting the feasibility and effectiveness of this strategy.

On the other hand, the PANGEA trial [66] made a step forward proposing a similar strategy but with an algorithm for the first three lines of treatment. Every line included a combination of chemotherapy plus a monoclonal antibody. The selected antibodies were target therapies against tumours expressing HER2, MET, FGFR2 or EGFR amplifications, MAPK/PIK3CA alterations, or immune deregulations (MSI, EBV, CPS ≥ 10 or TMB ≥ 15 mut/Mb). The molecular analyses performed in the PANGEA trial revealed genomic discordances between the primary and the metastatic lesion in 35% of patients, and for them treatments were chosen based on the metastatic lesion profile. This study was able to demonstrate the well-recognised spatial heterogeneity of GC, but also the hypothesised temporal heterogeneity. Tumour changes were observed after the first- and second-targeted line in up to 50% of. The OS was the primary end point of this study, which was met, thus demonstrating how this strategy is feasible and effective.

After the success of these trails, a large-scale sequencing strategy was proved in Japan. By the GI-SCREEN platform [67], GC patients were screened and those with potential molecularly targetable alterations were derived to matched clinical trials. Afterwards, the same investigators carried out the GOZILA initiative [68], a demographic initiative to perform a comprehensive ctDNA sequencing for all GC patients, to rapidly screen them for trial eligibility. This strategy aims to demonstrate the feasibility and efficacy of performing such strategies focused on a population level. Interestingly, they have shown preliminary data, including the possibility of detecting and offering targeted treatments to patients carrying rare molecular alterations such as NTRK1 fusions. 

## 5. Conclusions

GC accounts for the fifth most common tumour type and the fourth cause of cancer mortality [1], a worldwide healthcare problem that needs to be better addressed. Whether historically CG has been treated as a unique disease, molecular approaches have largely demonstrated its inherent spatial and temporal heterogeneity [6,8], and fortunately the paradigm of this disease is switching and considering this diversity.

The benefit of a systemic treatment (chemotherapy) in patients with locally advanced unresectable and/or metastatic GC has been robustly demonstrated, both in terms of OS and QoL [2]. Considering the aggressivity of this disease and the demonstrated preservation of the QoL by every treatment line, the minor gain derived from each treatment makes the quality adjusted life years (QALY) even higher than the demonstrated gain in the OS. After two therapy lines, a third line of treatment should be offered in those patients that maintains a good performance status and desire to continue. As further lines of treatment have been correlated with a better OS and with an improvement of the QoL, a close clinical follow-up for preserving PS, and an active and periodic radiologic assessment to detect disease progression at a very early stage is required [69]. A third line with a taxane or irinotecan, depending on the second line, is acceptable. Nevertheless, trifuridine/tipiracil is the only chemotherapy drug with positive results in a phase III trial [45], and it should, therefore, be considered when possible. Where targeted therapies have failed to demonstrate any efficacy in an unselected GC population, T-Dxd seemed active in the HER2-positive population [60]. On the other hand, immune combinations to reverse anti-PD1 resistances are being evaluated and will likely become part of the treatment armamentarium in the future. Apart from HER2- and PD-1 CPS-positive GC populations, MSI-H/dMMR status should be also considered as a biomarker of immune response [51], and these patients should be accordingly treated with ICIs.

Two major reasons have contributed to the failure of the main phase III trials in GC. On the one hand, related to the tumour. GC intrinsic spatial and temporal heterogeneity has historically impeded the identification of proper molecular aberrations. On the other hand, linked to clinical trial issues, mainly derived from a lack of biomarker-based stratification and/or insufficient novel designs. The recognition of these factors represents an important step forward. Now, better efforts should be made to reach a complete diagnose of the GC patient from the beginning, considering the GC patient as a whole entity, from the clinical point of view up to a deep molecular analysis of the tumour.

Hopefully, the GC treatment paradigm will soon improve. The integration of better follow-ups with multidisciplinary teams to attend patients’ symptoms, a more intensive radiologic assessment, and the more accurate molecular analysis, would derive on improved and personalized algorithms of treatment, from the very beginning of the disease, and would finally translate in a better survival and QoL of these patients. 

## Figures and Tables

**Figure 1 cancers-14-01408-f001:**
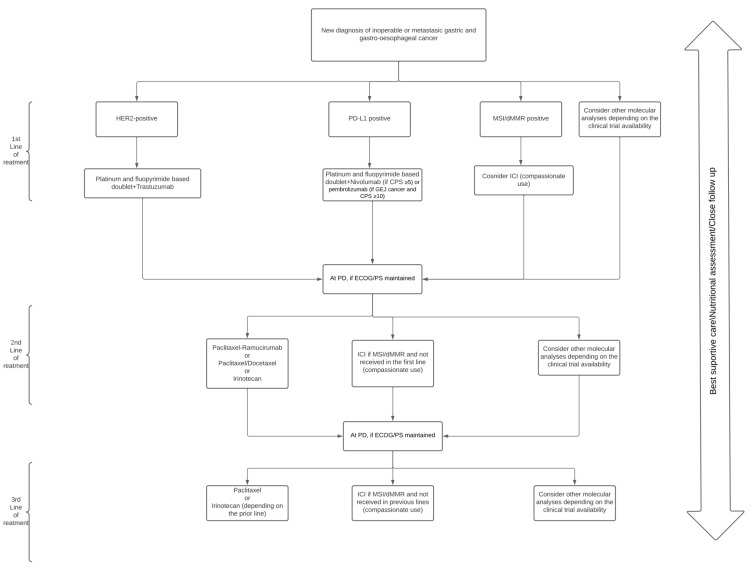
Treatment algorithm of patients with gastric and gastro-oesophageal junction cancer.

**Table 1 cancers-14-01408-t001:** Clinical trials in the refractory setting in patients with gastric and gastro-oesophageal junction cancer.

Clinical Trial	N	Regimen	OS	Hazard Ratio *p* Value	PFS	Hazard Ratio *p* Value	ORR	*p* Value
2 Line								
DESTINY- Gastric02 Van Cutsem et al. Ann Oncol 2021	79	T-DXd	-	-	5.5 m	-	38%	-
2-3 Line								
GRANITE-1 Trial Othsu et al. J Clin Oncol 2013	656	Eve PB	5.4 m 4.3 m	HR: 0.90 *p* = 0.124	1.7 m 1.4 m	HR: 0.66 *p* < 0.001	4.50% 2.10%	-
3 Line and beyond								
TAGS Trial Shitara et al. Lancet Oncol 2018	507	TAS-102 PB	5.7 m 3.6 m	HR: 0.69 *p* < 0.01	2.0 m 1.8 m	HR: 0.57 *p* < 0.01	4% 2%	*p* = <0.28
ATTRACTION-02 Kang et al. Lancet 2017	493	Nivo Placebo	5.3 m 4.1 m	HR: 0.63 *p* < 0.001	1.6 m 1.45 m	HR: 0.60 *p* < 0.001	11.20% 0%	-
KEYNOTE-059 Fuchs et al. JAMA Oncol 2018	259	Pem	5.6 m	-	2	-	11.60%	-
JAVELIN 300 Bang et al. Ann Oncol 2018	371	Ave CPT-11/Pac	4.6 m 5.0 m	HR: 1.1 *p* = 0.81	1.4 m 2.7 m	HR: 1.73 *p* > 0.99	2.2% 4.3%	-
Apatinib Trial Li et al. J Clin Oncol 2016	267	Apa PB	6.5 m 4.7 m	HR: 0.70 *p* = 0.015	2.6 m 1.8 m	HR: 0.44 *p* < 0.01	2.84% 0%	*p* = 0.169
ANGEL Trial Kang Y-K et al. Ann Oncol 2019	460	Apa PB	5.78 m 5.13 m	HR: 0.93 *p* = 0.4850	2.83 m 1.77 m	HR: 0.57 *p* < 0.0001	6.87% 0%	*p* = 0.0020
DESTINY- Gastric01 Shitara et al. N. Engl. J. Med 2020	188	T-DXd CPT-11/Pac	12.5 m 8.4 m	HR: 0.59 *p* = 0.01	5.6 m 3.5 m	HR: 0.47 -	51% 14%	*p* = <0.001

“–“, not reported; OS: Overall Survival; PFS: Progression Free Survival; ORR: overall response rate; HR: hazard ratio; m: months; TAS-102: trifluridine/tipiracil; PB: Placebo; Nivo: Nivolumab; Pem: pembrolizumab; Ave: Avelumab; Apa: Apatinib; Eve: Everolimus; T-DXd: trastuzumab deruxtecan; CPT-11: irinotecan; Pac: paclitaxel.

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
