# Peer review of "Chemorefractory Gastric Cancer: The Evolving Terrain of Third-Line Therapy and Beyond"

_cancers, 2022, doi:10.3390/cancers14061408_

Round 1

Reviewer 1 Report

This review help us to organize and understand the new treatments for the patients with chemorefractory gastric cancer.
However, if there is a table to refer to, it will be more helpful for reader to understand which biomarker can be used at what time and in what situation.

Author Response

We thank the reviewer for the comment.

We have modified and improved the fig. 1 to facilitate the comprehension of the manuscript, according to this reviewer comment.

Reviewer 2 Report

Great overall review. I enjoyed reading and thanks for sending for review.

Content recs:

  1. What are the specifics on who is fit for a second (or even first line) therapy? Are there objective data in patient selection? Probably over and under patient selection can contribute significantly to treatment outcomes - which the author touch on, but how to optimize who is treated is not covered in detail. Are we treating too many people to too few? 
  2. How do we differentiate distal esophageal cancer from proximal gastric cancer? Does it change treatment?
  3. Given the relatively small gains in OS, what was the cost/benefit in terms of QOL for treatment? It may be QALY are much less impacted (or potentially harmed) relative to OS if treatments decrease QOL during and after treatment. Alternatively, if QOL is improved, those gains may make QALY gains higher that OS and potentially provide for significant findings. 
  4. What about differences in East v. West - etiology, operative management, base patient phenotype and tumor phenotype differ, does this matter in selecting patients to treat or treatment options. Are outcomes different?

Editorial note:

1. Analysis was spelled analisis in figure one (not sure is this is UK spelling).

Author Response

Content recs:

What are the specifics on who is fit for a second (or even first line) therapy? Are there objective data in patient selection? Probably over and under patient selection can contribute significantly to treatment outcomes - which the author touch on, but how to optimize who is treated is not covered in detail. Are we treating too many people to too few?

We thank the reviewer for the comments. We have added a paragraph specifying this issue in the “close follow-up” section.

How do we differentiate distal esophageal cancer from proximal gastric cancer? Does it change treatment?

We thank the reviewer for this question, which is actually very important, mostly considering the trials with immunotherapy in the first line setting of gastric and oesophageal cancer. Nevertheless, we were asked to write a review of the treatment of gastric cancer, not oesophageal, thus not having able to discuss this issue.

Given the relatively small gains in OS, what was the cost/benefit in terms of QOL for treatment? It may be QALY are much less impacted (or potentially harmed) relative to OS if treatments decrease QOL during and after treatment. Alternatively, if QOL is improved, those gains may make QALY gains higher that OS and potentially provide for significant findings.

We agree with the reviewer. We have supported this comment by adding a paragraph within the conclusions.

What about differences in East v. West - etiology, operative management, base patient phenotype and tumor phenotype differ, does this matter in selecting patients to treat or treatment options. Are outcomes different?

Although the well described different prognosis of resectable GC patients depending on east vs west origin, data on the metastatic/refractory setting is controversial and not convincing. Molecular characterization seems to be the same and thus they should respond similarly. This article treats only the metastatic/refractory setting, and therefore we though this issue could be add confusion.

Editorial note:

Analysis was spelled analisis in figure one (not sure is this is UK spelling).

It has been changed.

Reviewer 3 Report

Your manuscript is interesting and well written about chemotherapy for gastric cancer. However, I think your manuscript needs further improving. 

#1. Please change the order of table 1 and the chemotherapy section according to therapeutic lines. E.g. first line, second line, and third lines. 

#2. I recommend adding further figures to show a molecular targeting of chemotherapy for gastric cancer.

#3. Gastrointestinal tract has a microbiota, not lung and kidney organs. Several bacteria of these gut microbiota have a potential carcinogenesis such as Helicobacter pylori. Thus, many readers are interested in the potential association between these bacteria and the molecular subtype of gastric cancer. Please add an additional section about that point.

Author Response

Your manuscript is interesting and well written about chemotherapy for gastric cancer. However, I think your manuscript needs further improving. 

#1. Please change the order of table 1 and the chemotherapy section according to therapeutic lines. E.g. first line, second line, and third lines. 

We thank the reviewer for this comment, and we have changed the order of table 1 according to first, second and third lines.

#2. I recommend adding further figures to show a molecular targeting of chemotherapy for gastric cancer.

We thank the reviewer for this comment and, also considering the reviewer 1 comment, we have modified fig. 1 to facilitate the comprehension of the required molecular tests in gastric cancer.

#3. Gastrointestinal tract has a microbiota, not lung and kidney organs. Several bacteria of these gut microbiota have a potential carcinogenesis such as Helicobacter pylori. Thus, many readers are interested in the potential association between these bacteria and the molecular subtype of gastric cancer. Please add an additional section about that point.

An additional section has been added to clarify the role of Helicobacter pylori and the microbiome in gastric cancer (introduction).